# The Loss of PPARγ Expression and Signaling Is a Key Feature of Cutaneous Actinic Disease and Squamous Cell Carcinoma: Association with Tumor Stromal Inflammation

**DOI:** 10.3390/cells13161356

**Published:** 2024-08-15

**Authors:** Raymond L. Konger, Xiaoling Xuei, Ethel Derr-Yellin, Fang Fang, Hongyu Gao, Yunlong Liu

**Affiliations:** 1Department of Pathology & Laboratory Medicine, Richard L. Roudebush Veterans Affairs Medical Center, Indianapolis, IN 46202, USA; 2Department of Pathology & Laboratory Medicine, Indiana University School of Medicine, Indianapolis, IN 46202, USA; ederryel@iu.edu; 3Department of Medical & Molecular Genetics, Indiana University School of Medicine, Indianapolis, IN 46202, USA; xxuei@iu.edu (X.X.); ffang@indiana.edu (F.F.); hongao@iu.edu (H.G.); yunliu@iu.edu (Y.L.)

**Keywords:** peroxisome proliferator-activated receptor, inflammation, tumor suppression, non-melanoma skin cancer, actinic keratoses, cutaneous squamous cell carcinoma, transcriptomics, gene set enrichment analysis

## Abstract

Given the importance of peroxisome proliferator-activated receptor (PPAR)-gamma in epidermal inflammation and carcinogenesis, we analyzed the transcriptomic changes observed in epidermal PPARγ-deficient mice (*Pparg*-/-^epi^). A gene set enrichment analysis revealed a close association with epithelial malignancy, inflammatory cell chemotaxis, and cell survival. Single-cell sequencing of *Pparg*-/-^epi^ mice verified changes to the stromal compartment, including increased inflammatory cell infiltrates, particularly neutrophils, and an increase in fibroblasts expressing myofibroblast marker genes. A comparison of transcriptomic data from *Pparg*-/-^epi^ and publicly available human and/or mouse actinic keratoses (AKs) and cutaneous squamous cell carcinomas (SCCs) revealed a strong correlation between the datasets. Importantly, PPAR signaling was the top common inhibited canonical pathway in AKs and SCCs. Both AKs and SCCs also had significantly reduced *PPARG* expression and PPARγ activity z-scores. Smaller reductions in *PPARA* expression and PPARα activity and increased *PPARD* expression but reduced PPARδ activation were also observed. Reduced PPAR activity was also associated with reduced PPARα/RXRα activity, while LPS/IL1-mediated inhibition of RXR activity was significantly activated in the tumor datasets. Notably, these changes were not observed in normal sun-exposed skin relative to non-exposed skin. Finally, *Ppara* and *Pparg* were heavily expressed in sebocytes, while *Ppard* was highly expressed in myofibroblasts, suggesting that PPARδ has a role in myofibroblast differentiation. In conclusion, these data provide strong evidence that PPARγ and possibly PPARα represent key tumor suppressors by acting as master inhibitors of the inflammatory changes found in AKs and SCCs.

## 1. Introduction

PPARs are members of a large family of related ligand-activated nuclear receptors that bind to a common consensus recognition sequence in the promoters of target genes, the peroxisome proliferators response element (PPRE). The ligand-induced activation of PPARs bound to the PPRE subsequently induces target gene transcription (for review, see [1]). Three different PPAR proteins have been identified (PPARα, PPARδ, and PPARγ), all of which require heterodimerization with the retinoid X receptor α (RXRα) for PPRE binding and transcriptional activity [1]. Endogenous PPAR ligands include unsaturated and saturated fatty acids and fatty acid metabolites [2,3]. While the three different PPAR receptors exhibit differences in tissue distribution, all three PPARs are known to be expressed in human and mouse skin [4]. In addition, all three PPAR subtypes play key roles as homeostatic regulators of lipid metabolism, energy balance, and cellular differentiation [2,3]. 

In addition to its role as a direct transcriptional regulator of genes involved in energy balance and lipid metabolism, PPARγ acts through mechanistically distinct transrepressive signaling pathways to suppress the activities of other transcription factors, such as NF-κB, activator protein 1 (AP-1), and the nuclear factor of activated T cells (NFAT) [5]. Using a whole transcriptomic analysis of differentially expressed genes in *Pparg-/-*^epi^ relative to wildtype control mice, we showed that *Pparg-/-*^epi^ mice exhibit a marked increase in inflammatory mediators and gene products associated with inflammasome activation, indicating that PPARγ has a key role as an important immune modulator [6]. The mice also developed spontaneous inflammatory skin lesions [6]. PPARγ was seen to have a role as a suppressor of cutaneous inflammation in studies of inflammatory dermatoses. Relative to normal human control skin, PPARγ transcripts in psoriatic and atopic lesions are reduced by 8- and 3.3-fold, respectively [7]. Another study demonstrated that *PPARG* mRNA is significantly decreased in human lichen planopilaris, a form of scarring (cicatricial) alopecia [8]. 

PPARs have also received interest for their potential role in neoplastic development. The keratinocyte-specific loss of *Pparg* in mice *(Pparg-/-*^epi^ mice) results in increased photocarcinogenesis and photoinflammation [9], as well as a severe defect in normal contact hypersensitivity (CHS) responses [10]. Similarly, the epidermal-specific loss of either *Pparg* or its heterodimeric partner RXRα in mice resulted in an approximately 2-fold increase in DMBA/PMA-induced tumors [11]. DMBA treatment in mice with hemizygous germline loss of *Pparg* resulted in a cancer incidence increase of over 3-fold, while metastatic disease increased by 4.6-fold [12]. The increase in cancers included a 1.7-fold increase in cutaneous papilloma multiplicity. 

In addition to these loss-of-function models, pharmacologic gain-of-function studies also indicate the potential role of PPARγ agonists in cutaneous malignancy. The treatment of mice with rosiglitazone was shown to suppress chemical carcinogenesis by approximately 70% [13]. Rosiglitazone also blocks the ability of ultraviolet light to suppress both CHS responses and anti-tumor immunity [10]. In addition, rosiglitazone promotes anti-tumor immune reactions in a mouse immunogenic cutaneous SCC tumor model [14]. In yet another study, the anti-neoplastic efficacy of immune therapy consisting of CTLA4 blockade and a cancer vaccine was enhanced by the addition of rosiglitazone treatment [15]. Collectively, these data indicate a potential tumor suppressor role of PPARγ that is mediated by its effects on inflammation and anti-tumor immune responses. 

Chronic inflammation and immunosuppression are hallmarks of the tumor microenvironment [16,17]. While malignant tumors orchestrate changes to the stromal microenvironment to promote tumor growth and escape from immune surveillance, it is unclear how this process is regulated. Mice lacking epidermal *Pparg* (*Pparg*-/-^epi^ mice) also exhibit immune suppression, chronic inflammation, and increased chemical and photocarcinogenesis. We were therefore interested in determining whether the transcriptomic changes observed in *Pparg*-/-^epi^ mice exhibit any overlap with non-melanoma skin cancer (NMSC). We utilized a transcriptomic analysis and single-cell RNA sequencing (scRNAseq) to provide evidence that PPAR signaling is the top inhibited canonical signaling pathway in NMSC and that this loss correlates with significant reductions in PPARγ expression and activity and increased cytokine and chemokine signaling. Smaller but significant reductions in PPARα mRNA expression and activity were also seen, suggesting a potential tumor suppressor role for PPARα as well. In *Pparg*-/-^epi^ mice, scRNAseq further indicated that epidermal *Pparg* is a key epidermal regulator that modulates the recruitment of stromal myeloid, lymphoid, and fibroblast populations. 

## 2. Materials and Methods

### 2.1. Animal Studies

The derivation of mice lacking epidermal *Pparg* in the C57BL/6 background (*Pparg*-/-^epi^) and wildtype controls (lacking Cre recombinase) was previously described [10]. Mice were housed under specific pathogen-free conditions at the Indiana University School of Medicine. 

### 2.2. Whole Transcriptomic mRNA Sequencing

mRNA sequencing and a subsequent differentially expressed gene analysis were performed on epidermal scrapings of *Pparg*-/-^epi^ mice relative to wildtype controls as previously described [6]. Sequencing was carried out on RNA obtained from the skin of 6 mice per genotype [6]. In this previously published study, we characterized how changes in gene expression correlated with the phenotypic changes that we observed in *Pparg*-/-^epi^ mice. This included spontaneous inflammatory skin lesions, an asebia phenotype, and epidermal permeability defects. This dataset is also publicly available at the Gene Expression Omnibus (GEO), National Center for Biotechnology Information (NCBI) depository (accession number: GSE164024). 

### 2.3. Single Cell Isolation

Two mice (one male and one female) were euthanized for each genotype (wildtype and *Pparg*-/-^epi^). Dermal and epidermal cells were isolated from areas of telogen phase hair cycling for each mouse. For dermal cell preparation, the tissues were minced and digested in 2.5 mg Liberase TM, 16.7 mg DNase I, penicillin, and streptomycin in RPMI media (all reagents from Sigma-Aldrich, St. Louis, MO, USA) for 90 min at 37 °C. The digest was filtered through a 40 μm filter. After adding 5 mL of ice-cold dialyzed FBS, the samples were centrifuged at 300× *g* for 10 min. RBC lysis (RBC Lysis Buffer, Cat #00-4333-57, ThermoFisher Scientific, Waltham, MA, USA) followed by dead cell removal (dead cell removal kit #130-090-101, Miltenyi Biotec, Auburn, CA, USA) with centrifugation were performed after each step. The cell prep was washed twice with cell suspension buffer [PBS (calcium, magnesium, DNase, and RNase free) (Catolog # 40120706, bioWORLD, Dublin, OH, USA) containing 1% BSA] followed by centrifugation. RNA sequencing was then performed on a final cell prep resuspended in cell suspension buffer containing 0.5 U/µL SUPERaseIn™ RNase Inhibitor (Catalog #AM2694, ThermoFisher Scientific, Waltham, MA, USA) and 0.5 U/µL Protector RNase Inhibitor (Catalog #3335399001, Sigma-Aldrich, St. Louis, MO, USA). 

For epidermal cell preps, the dermal fat and dermis were removed using a scalpel, and the epidermal sheet was suspended in dispase II solution Sigma-Aldrich, St. Louis, MO, USA) for 2 hrs at 37 °C. After suctioning the dispase and scraping the epidermis from the dermis, the tissue was rinsed with PBS, centrifuged, and incubated in 5 mL Accutase solution (Cat# CnT-Accutase-100, CELLnTEC Advanced Cell Systems, Bern, Switzerland) for 20 min at room temp. After filtering through a 70 µm filter, 5 mL of dialyzed FBS was added, and the cells were pelleted by centrifugation. Dead cell removal and further processing were then performed as described for the dermal cell prep. The epidermal cells and dermal cells were then recombined at a 1:10 ratio (dermal vs. epidermal), and the male and female cells were pooled together. 

### 2.4. Single-Cell Sequencing and Library Prep Protocol

After evaluating for cell number, cell viability, and cell size, the appropriate number of cells were loaded on a multiple-channel micro-fluidics chip of the Chromium Single-Cell Instrument (10x Genomics, Pleasanton, CA, USA) with a targeted cell recovery of 10,000. Single-cell gel beads in emulsion containing barcoded oligonucleotides and reverse transcriptase reagents were generated with the v3.1 Next GEM Single Cell 3′ reagent kit (10X Genomics, Pleasanton, CA, USA). Following cell capture and cell lysis, cDNA was synthesized and amplified. An Illumina sequencing library was then prepared with the amplified cDNA with the Chromium™ Next GEM Single-Cell 3′ GEM, Library & Gel Bead Kit v3.1 (10X Genomics, Pleasanton, CA, USA). The resulting library was sequenced, including cell barcode and UMI sequences, and 100 bp RNA reads were generated with Illumina NovaSeq 6000 (Illumina, San Diego, CA, USA) at the Center for Medical Genomics of the Indiana University School of Medicine.

### 2.5. Analysis of scRNAseq Data

CellRanger 6.1.1 (http://support.10xgenomics.com/ 10X Genomics, Pleasanton, CA, USA. Accessed on 7 January 2022) was utilized to process the raw sequence data generated. Briefly, cellranger mkfastq was implemented to demultiplex raw base sequence calls generated from the Illumina sequencer into sample-specific FASTQ files. The FASTQ files were then aligned to the mouse reference genome mm10 with RNAseq aligner STAR. The aligned reads were traced back to individual cells, and the gene expression level of individual genes were quantified based on the number of UMIs (unique molecular indices) detected in each cell. The filtered feature-cell barcode matrices generated using CellRanger were used for further analysis. 

### 2.6. SoupX Analysis

The R package SoupX version 1.5.2 [18] was used to remove cell-free contaminating RNA from the data.

### 2.7. Seurat Analysis

The R package Seurat version 4.0 [19,20,21,22] was used for the following analyses: cell type/state discovery with graph-based clustering, cell cluster marker gene identification, and various visualization. A QC metrics of the library size, number of features/genes, and mitochondrial reads (based on median absolute deviation (MAD); MAD of 3 used here) was calculated with Scater [23]. This, together with the QC analysis in Seurat, were used to determine the parameters used for excluding low-quality cells.

### 2.8. Performing QC and Selecting Cells for Further Analysis

Low-quality cells were excluded with the following criteria: cells with unique feature/gene counts over 7000 or less than 300 or >10% reads mapped to mitochondrial genome. A summary of the final scRNAseq data output is shown in Appendix A (WT mouse cells) and Appendix A (*Pparg*-/-^epi^ mouse cells). 

### 2.9. Unsupervised Cell Cluster Segregation and Cluster Identification

Unsupervised clustering was performed using Loupe Browser software (v6.5.0; 10x Genomics, Pleasanton, CA, USA) by uniform manifold approximation and projection (UMAP). Using Loupe Browser, DEGs for each individual cell cluster were obtained. The cell type within each cluster was identified by uploading the top differentially expressed genes for each cluster into the Enrichr online analysis application [24,25,26] and then cross-referencing with the PanglaoDB database [27]. 

### 2.10. GTEX Data Analysis

Tissue-specific (sun-exposed lower leg skin and sun-protected suprapubic skin) TPM for *PPARA*, *PPARD*, and *PPARG* were downloaded from the GTEX portal [www.gtexportal.org]. After uploading the TPM data, differential expression for the PPARs was calculated using edgeR (DEApp website: yanli.shinyapps.io/DEApp/. Accessed 6 December 2023) [28].

### 2.11. Gene Set Enrichment Analysis

Ingenuity Pathway Analysis (IPA) was performed to identify predicted diseases and biofunctions, canonical pathways, and upstream regulators that are enriched in datasets relative to what would be expected by chance (QIAGEN Inc., Germantown, MD, USA. https://digitalinsights.qiagen.com/IPA, Accessed 6 December 2023) [29]. 

## 3. Results

### 3.1. The Transcriptomic Changes in Pparg-/-^epi^ Mice Show a Strong Correlation to Inflammatory Cell Recruitment and Mobilization

To better understand how the loss of epidermal *Pparg* reflects disease processes, we reanalyzed our transcriptomic dataset from *Pparg*-/-^epi^ mouse skin and wildtype mouse skin (GSE164024, [6]). After obtaining the list of differentially expressed genes (DEGs) that we observed in *Pparg*-/-^epi^ mouse skin relative to wildtype mouse skin, we uploaded this dataset into Ingenuity Pathway Analysis (IPA, Qiagen, Germantown, MD, USA) and then performed a gene set enrichment (GSE) analysis for diseases and biofunctions (the complete annotated list is in Appendix A). 

A potential clue of how PPARγ could influence cancer is seen in Table 1, in which the top 20 matching disease and biofunction terms are shown after sorting by the highest positive activation z-scores. All 20 had activation z-scores of 3.942 to 5.052, well above the 2.0 cutoff that is used to predict activation. Of these, 15 of the biofunctions were associated with inflammatory cell homing, chemotaxis, or the inflammatory response. The remaining biofunctions were related to cell viability, tumor or cell invasion, and the growth of lesions or tumors. Given the important role of inflammation in cancer, these data suggest that the primary interaction of PPARγ in tumor development may center on its anti-inflammatory activity. Overall, this approach found a strong linkage to inflammatory biofunctions, as shown in Table 1. 

### 3.2. Single-Cell Sequencing of Pparg-/-^epi^ Mice Reveals Increase in Immune Cells, Particularly Neutrophils

To determine how the loss of epidermal *Pparg* alters the stromal immune environment, we next performed single-cell sequencing of skin from both the WT and *Pparg*-/-^epi^ mice. We found that 1560 genes were differentially expressed in the *Pparg*-/-^epi^ cells relative to the WT cells (Figure 1A; a full list of DEGs can be found in Appendix A). Of these, 94.9% of the DEGs were shared with the DEGs found in our previous whole transcriptomic RNAseq dataset [6]. After an unsupervised cluster analysis, 26 different cell clusters were identified (Figure 1B). Appendix A shows the Enrichr analysis for the cell types found in each cell cluster. After focusing on non-cytokeratin-expressing cells, we found that 44.11% of non-keratinocytes represented *Ptprc* (CD45)-expressing immune cells in *Pparg*-/-^epi^ mouse skin (Figure 1C and Appendix A). In contrast, 20.42% of non-keratinocytes were *Ptprc*^+^ in the WT mouse skin (Figure 1C and Appendix A). This increase in immune cells in *Pparg*-/-^epi^ mouse skin relative to other stromal cell types was reflected as an increase in both myeloid and lymphoid cell populations (Figure 1C). 

After a further analysis of only the myeloid cell populations, we found that neutrophils represented only 0.6% of the total myeloid cell population observed in the WT mice (Figure 1D and Appendix A). This low neutrophil count is not particularly surprising as neutrophils are infrequent in normal mouse skin [30]. In contrast, neutrophils represented 25.93% of the total myeloid cell population in *Pparg*-/-^epi^ mice, which is a 43-fold increase relative to the WT mice (Figure 1D). 

### 3.3. Fibroblasts Expressing Myofibroblast Markers Are Increased in Pparg-/-^epi^ Mouse Skin

In addition to differences in immune cell clusters, we also found differences in the stromal fibroblasts from *Pparg*-/-^epi^ mouse skin. In Figure 2A, we show the expression of the collagen type I alpha 2 gene (*Col1a2*) in non-immune stromal cell clusters. Clusters 16 and 20, along with clusters 1, 3, 5, 9, 10, and 21, expressed high levels of *Col1a2*, consistent with the Enrichr identification of these clusters as fibroblasts. Also as expected, *Col1a2* gene expression was absent in clusters representing primarily melanocytes and mast cells (cluster 4) and smooth muscle cells (cluster 18). 

The Enrichr analysis indicated that clusters 16 and 20 have differential gene expression overlap with pancreatic stellate cells (PSCs) (Appendix A). As activated PSCs are highly fibrogenic myofibroblasts (myoFBs) [31], we verified that these fibroblast clusters express myofibroblast-specific genes. Figure 2B,C show the expression of S100 calcium binding protein A4 (*S100a4*) [also known as fibroblast specific protein 1] and alpha smooth muscle actin (*Acta2*), respectively. Both *S100a4* and *Acta2* are expressed in mouse dermal myofibroblasts, although *Acta2* is the more specific marker [32]. Only fibroblast clusters 16 and 20 expressed both markers, with the expression being particularly high in cluster 16. As an internal positive control, *Acta2* was highly expressed in the smooth muscle cell cluster (cluster 18). 

In Figure 2D, we show that cluster 16 myofibroblasts were infrequent in the WT dermis (0.28% of non-immune stromal cells). In contrast, cluster 16 myofibroblasts were enriched in *Pparg*-/-^epi^ mice, representing nearly 5% of non-immune stromal cells (relative fold change of 17.7-fold over WT mice). Similarly, cluster 20 myofibroblasts were enriched by nearly 4-fold in *Pparg*-/-^epi^ mouse skin (26.34% vs. 6.68% of non-immune stromal cells in *Pparg*-/-^epi^ and WT skin, respectively). 

### 3.4. The GSE Analysis Reveals Strong Similarity between the Pparg-/-^epi^ Transcriptomic Data and That of Human Actinic Disease and Mouse and Human SCCs

To verify that the chronic inflammatory microenvironment observed in *Pparg*-/-^epi^ mice is relevant to non-melanoma skin cancer, we uploaded DEG datasets for both our whole transcriptomic RNAseq [6] and the single-cell RNAseq (Appendix A) for the IPA. We also uploaded DEGs for publicly available transcriptomic datasets of human actinic keratoses (AKs), human SCCs, and mouse SCCs (relative to normal skin). (Details of the publicly available datasets can be found in Appendix A.) To determine how closely the *Pparg*-/-^epi^ datasets correlate with the different tumor datasets, we performed a comparison analysis of diseases and biofunctions. 

Figure 3A,B show heatmaps that demonstrate the different disease and biofunction processes that are significantly associated with the datasets, and they are sorted by z-score. Figure 3A is limited to a comparison between *Pparg*-/-^epi^ mouse skin data and human AK or SCC datasets, while Figure 3B compares *Pparg*-/-^epi^ data to that of mouse SCC. Both the whole transcriptomic and single-cell RNA sequencing datasets for *Pparg*-/-^epi^ mice were largely in agreement, with the observed differences generally being associated with the non-informative z-scores (z-score near 0) in the scRNAseq dataset due to the much smaller DEG profile for this less sensitive technique. For the tumor datasets, the activating scores varied but were largely in agreement across datasets. As with the two different *Pparg*-/-^epi^ datasets, the greatest differences across the different tumor datasets were largely due to non-informative terms. In addition, no consensus was seen for “Organismal death”, which was predicted to be activated, inhibited, or not effected depending on the dataset. Surprisingly, the correlation between *Pparg*-/-^epi^ mouse skin and human AKs and SCCs (Figure 3A) was stronger than that between *Pparg*-/-^epi^ mice and mouse cutaneous SCCs (Figure 3B). When the *Pparg*-/-^epi^ mouse dataset was compared to that of mouse SCCs, there were more non-informative functions observed in the *Pparg*-/-^epi^ dataset compared to the mouse tumor datasets. 

Given that *Pparg*-/-^epi^ mice are highly susceptible to cutaneous carcinogenesis, it is particularly interesting that there was close agreement between the *Pparg*-/-^epi^ transcriptomic datasets with the tumor datasets. In both Figure 3A,B, the activating z-scores for functional annotations associated with cell movement, chemotaxis, and cell invasion were strongly associated with both the *Pparg*-/-^epi^ and tumor datasets. Other functions that were activated in most datasets included various functions associated with cell viability, tumor growth, tumor invasion, and metastasis. 

BCCs represent the most common NMSC observed in humans. We therefore also ran a correlation GSE analysis of our *Pparg*-/-^epi^ datasets with six human BCC transcriptomic datasets (Appendix A). As BCCs are not found in mice lacking disruptions in patched signaling, it is not surprising that there was a limited correlation between the diseases and biofunction annotations that were common to both the *Pparg*-/-^epi^ dataset and the BCC datasets. However, in contrast to the AK and SCC datasets, there was considerable discordance for the disease and biofunctions linked to different BCC datasets. This could indicate a greater degree of tumor heterogeneity in BCCs. Poor predictive modeling could also be due to a reduced number of gene expression changes or more modest effect size changes in the DEGs associated with indolent BCCs. 

### 3.5. The Canonical Pathway Analysis Shows that the Loss of PPAR Signaling Is a Top Inhibited Canonical Pathway in Human AKs, Human SCCs, and Mouse SCCs

To determine whether similarities in diseases and biofunctions are associated with common signaling pathways, we utilized an IPA to compare the canonical pathways that are predicted to be activated or inhibited based on the DEGs found in *Pparg*-/-^epi^ mouse skin and those from human AKs or SCCs (Figure 4A), as well as a comparison with both human and mouse SCCs (Figure 4B). As with diseases and biofunctions, there was a relatively high level of consistency in the z-scores between the pathways mapped to *Pparg*-/-^epi^ mouse skin and human tumors (Figure 4A). Canonical pathways with largely positive z-scores were heavily associated with cytokine and chemokine signaling, innate and adaptive immune responses, pyroptosis, the tumor microenvironment, and fibrosis. These changes are not surprising given the known role of inflammation in tumor–stroma interactions. 

The strong correlation between the predicted activation of these pathways in both malignancy and *Pparg*-/-^epi^ mouse skin further supports the potential role of PPARγ as a tumor-suppressing signal through its anti-inflammatory activity. It is therefore of interest that “PPAR Signaling” was the top common canonical pathway, which shows largely negative z-scores in Figure 4A. Interestingly, the canonical pathway “LPS/IL-1 Mediated inhibition of RXR Function” was predicted to be activated in most of the AK and SCC datasets. This is of interest as the inhibition of RXRα activity could indirectly impact PPAR signaling as all PPARs require heterodimerization with RXRα for their transcriptional activity. In addition, as noted above, LPS and IL1β signaling also directly inhibit *PPARG/Pparg* transcript expression and target gene expression [33]. Thus, the activation of this pathway can suppress PPARγ signaling both directly and indirectly. 

In Figure 4B, we show a similar heat map that illustrates common canonical pathways that are activated or inhibited in *Pparg*-/-^epi^ mouse skin and either human or mouse SCCs. As with Figure 4A, the canonical pathway “PPAR Signaling” showed largely negative z-scores in all SCC datasets. In addition, several additional canonical pathways linked to PPAR signaling were also inhibited. The “PPARα/RXRα activation” and “LXR/RXRα activation” canonical pathways were also predicted to be inhibited in *Pparg*-/-^epi^ mouse skin and the tumor datasets. The predicted suppression of signaling by these two heterodimeric partners of RXRα is consistent with the predicted activation of the “LPS/IL-mediated inhibition of RXR signaling” that we observed in the human AK and SCC datasets (black arrows in Figure 4A,B). 

As with the diseases and biofunctions in Figure 3, there was more disagreement in the canonical pathways between the *Pparg*-/-^epi^ mouse skin RNAseq data relative to the mouse SCCs than there was between the *Pparg*-/-^epi^ mice and human SCCs (compare Figure 4A,B). These differences included positive z-scores for canonical pathways such as “Phagosome Formation”, “Role of NFAT in Regulation of the Immune Response”, “T cell receptor signaling”, and “FAK Signaling” for *Pparg*-/-^epi^ mice and human SCCs, but there was a trend of negative z-scores for these canonical pathways in mouse SCCs. 

Finally, we also examined canonical signaling pathways in human BCCs (Appendix A). As with the diseases and biofunctions analysis, canonical signaling pathways that were annotated to the *Pparg*-/-^epi^ mouse or BCC datasets showed a poor consensus. This suggests that the gene expression changes that occur with the loss of epidermal *Pparg* may not be particularly relevant to the top canonical signaling pathways that are observed in BCCs. 

### 3.6. A Shift to Reduced PPAR and RXR Activity and Reduced PPARA/Ppara and PPARG/Pparg Expression but Increased PPARD/Ppard Expression Occurs during the Progression from Sun-Exposed Skin to NMSC

Given that PPAR and RXR signaling represent the top inhibited canonical signaling pathways in Figure 4A,B, we further examined how PPAR signaling, PPARα/RXRα activation, and the LPS/IL1-mediated inhibition of the RXR function were altered during tumor progression. 

In Figure 5A, we plot the mean z-scores for the canonical pathway “PPAR Signaling” for the human and mouse tumor dataset as well as human sun-exposed skin (hSES). For hSES relative to non-sun-exposed skin (NES), “PPAR Signaling” had significantly positive mean z-scores near the z-score activation cutoff of 2.0. In contrast, the mean z-scores were significantly reduced for human AKs, human SCCs, and mouse SCCs. In all three cases, the means were near the inhibitory z-score threshold of −2.0. The shift from a positive “PPAR Signaling” z-score in SES to a negative z-score in human SCC was also statistically significant. 

In Figure 5B, the mean z-scores for the canonical pathway “PPARα/RXRα Activation” are similarly shown. In the case of SES vs. NES, the datasets were non-informative, indicating no impact on the canonical pathway “PPARα/RXRα activation”. However, negative mean z-scores were observed for human AK, human SCC, and mouse SCC. Again, the mean z-score for human BCCs was not significantly different from zero. 

Figure 5C shows the z-score distribution for the different tumor types when the canonical pathway “LPS/IL1-mediated inhibition of RXR function” was plotted. Again, the SES vs. NES datasets were non-informative. For human AKs, human SCCs, and mouse SCCs, the mean z-scores were near or exceeded the activation cutoff of 2.0. In contrast to Figure 5A,B, human BCCs showed a significantly positive mean z-score, although this mean score was well below the activation cutoff. 

As PPAR isoforms have significant overlap in target genes, the decrease in the canonical pathway PPAR signaling that is seen in Figure 5A could reflect changes in the activity of one or more of the PPARα, PPARδ, or PPARγ isoforms. Moreover, the decrease in PPARα/RXRα the canonical pathway’s activation could indicate a reduction in PPARα activation. Alternatively, the increased activation of the canonical pathway “LPS/IL1-mediated inhibition of RXR function” indicates that RXRα activity is likely reduced, a result that would impact the activity of all three PPAR isoforms. We therefore examined PPAR isotype transcript expression in human sun-exposed skin (SES), AKs, human SCCs, and mouse cutaneous SCCs (Figure 5D).

In Figure 5D, the mean *PPARG* expression was increased in SES relative to NES by approximately 1.5-fold. In the premalignant AK and malignant BCC and human SCC datasets, this was reversed: the mean *PPARG* expression was reduced by 39.6% in AKs, 62.3% in SCCs, and 84.0% in BCCs. This change from non-malignant SES was statistically significant for both the SCC and BCC datasets. In mouse SCCs, *Pparg* expression was also decreased by 84.7% relative to normal skin. The data for *PPARA/Ppara* in Figure 5D were similar in pattern to that observed for *PPARG/Pparg*, although to a lesser degree. However, the reduction in *PPARA* expression for the tumor datasets was not significantly different from that obtained from the SES datasets. In contrast to *PPARA* and *PPARG*, *PPARD* expression was not altered in SES or BCCs, but *PPARD/Ppard* was significantly increased in AKs and both human and mouse SCCs. 

A limitation of the analysis in Figure 5D is the relatively small number of studies that reported *PPAR* expression in SES vs. NES. No PPAR expression data were reported in the studies by Kita et al. [34] and Zou et al. [35] (Appendix A). However, both studies limited their reported DEGs to those that met both a statistical threshold and a fold change threshold. The absence of all three PPARs in their listed DEGs indicates that PPARs were only modestly or non-significantly altered in SES vs. NES. This idea was further supported by obtaining the GTEX dataset for SES (lower leg) and NES (suprapubic) and performing a differential expression analysis for the PPARs (Appendix A). As expected, the changes in *PPAR* expression were modest and none were significant. Thus, a significant alteration in *PPAR* expression is not likely a feature of chronically sun-exposed skin. Thus, a shift to reduced *PPARA* and *PPARG* expression is a feature of premalignant AKs as well as the two most common NMSCs (BCC and SCC). For *PPARD*, this shift is reversed for AKs and SCCs, with a significant increase in expression relative to normal skin. 

Interestingly, when we examined *Ppara* and *Ppard* expression in our *Pparg*-/-^epi^ whole transcriptomic dataset, we found that the loss of epidermal *Pparg* resulted in a 60% decrease in *Ppara* expression (FC = −2.482 (FDR = 1.89 × 10^−12^)) and a 1.5-fold increase in *Ppard* expression (FDR = 2.66 × 10^−4^) [6]. It is important to note that in *Pparg*-/-^epi^ mice, the observed changes to *Ppara* expression in the skin are triggered by the loss of *Pparg* only within keratinocytes. 

In the tumor datasets, it is unclear whether the observed changes in *PPAR* expression are occurring in the tumor epithelium or stromal cells. To address this question, *PPAR* isotype expression was obtained from a published dataset created by Mitsui et al. [36]. This study obtained transcriptomic data from AKs and SCC but utilized laser capture microdissection to limit the analysis to the epithelial tissue (Appendix A). In this case, the pattern of *PPAR* expression was similar to the observed changes in Figure 5D. *PPARA* expression was reduced by 66.6% and 65.7% in SCCs and AKs, respectively (Appendix A). *PPARG* expression was reduced by 58.3% in SCCs and 50.5% in AKs (Appendix A). Finally, *PPARD* expression was increased by 2.83-fold in SCCs and 2.48-fold in AKs (Appendix A). Thus, these data suggest that alterations in PPAR expression are likely observed in tumor cells themselves. Whether a similar alteration in expression occurs within stromal cells is unclear and requires additional studies. 

Since *PPAR* expression at the transcript level does not necessarily imply PPAR activity, we next performed an upstream regulator analysis via IPA to determine whether alterations in PPAR isoform activity occur in NMSC. In Figure 5E, we plot the predicted mean activation z-scores for PPARα, PPARδ, and PPARγ for the tumor datasets. In SES relative to NES, the mean z-scores for all three PPARs were not significantly different from zero, indicating that none of the PPARs are predicted to be activated or inhibited in normal chronically sun-damaged skin. In contrast, a trend towards decreased PPARα and PPARγ z-scores is seen in AK lesions, while significantly negative z-scores are observed in human SCCs and BCCs and mouse SCCs. In the cases of all three malignancies, the mean z-scores met or exceeded the z-score cutoff of −2.0, which predicted that both PPARα and PPARγ signaling are inhibited and correlate well with the expression data observed in Figure 5D. 

While the predicted activity for both PPARα and PPARγ match up well with the expression data, the activity score for PPARδ in Figure 5E is the opposite of what would be predicted simply by assessing *PPARD* expression in Figure 5D. This is particularly notable for the mouse SCCs in which *PPARD* expression is significantly elevated, while PPARδ activity is predicted to be inhibited. The discordant results between *PPARD* expression and PPARδ activity could potentially be explained by reduced RXRα activity. Thus, the inhibition of the RXR function, such as through LPS or IL1 signaling, could indirectly suppress all RXRα heterodimeric partners, including PPARδ. The idea that all heterotrimeric partners of RXRα might be inhibited is supported by a shift from increased “LXR/RXR Activity” canonical pathway z-scores in sun-damaged skin to a reduction in “LXR/RXR Activity” z-scores for AKs, BCCs, and human and mouse SCCs (Appendix A). 

Finally, conflicting data are seen in BCCs. Figure 5A,B indicate that PPAR signaling overall and PPARα/RXRα activation are not significantly altered in BCCs. Yet, like AKs and SCCs, *PPARA* and *PPARG* expression are decreased in Figure 5D, and the activity of all three PPARs are predicted to be inhibited in Figure 5E. While it is difficult to explain these discrepant results, it is possible that the differential epigenetic regulation of PPAR-specific target genes in neoplastic basal cell populations results in a skewed analysis of PPAR activity. 

### 3.7. Increased PPARD Expression Represents a Marker of Myofibroblast Populations Found in Pparg-/-^epi^ Skin

Given that the loss of PPAR signaling and *PPAR* expression is a feature of AKs and SCCs, we examined the expression of the three PPAR isoforms within the individual cell clusters of our scRNAseq dataset. In Figure 6A,B, *Ppara* and *Pparg* are highly expressed only within the small sebocyte cluster (cluster 19). This is not surprising as PPARα and PPARγ are key to the lipogenesis that is needed to produce the sebum present in mature sebocytes [37]. The importance of PPARγ in sebocyte differentiation is seen in the absence of lipid-filled sebocytes in the dermis of *Pparg*-/-^epi^ mice [6]. It is therefore not surprising that all of the sebocytes that were identified in the scRNAseq unsupervised cell clustering were found in the WT mice (Appendix A).

In contrast to *Ppara* and *Pparg, Ppard* was highly expressed only in the fibroblast clusters with myofibroblast features as well as the smooth muscle cell cluster (Figure 6C). This is somewhat surprising and suggests that high *Ppard* expression in mouse fibroblasts may represent a marker of myofibroblast differentiation. It might be noted that myofibroblasts are increased in the tumor microenvironment [38]. Thus, future studies are needed to determine whether cancer-associated fibroblasts with myofibroblast features also overexpress *Ppard*. As noted above, laser capture microdissection data of human AKs and SCCs indicate that increased *PPARD* occurs within the tumor cells themselves (Appendix A and [36]). Thus, it remains to be seen whether tumor-associated myofibroblasts may be contributing to the increase in *PPARD* mRNA that was observed in the NMSCs. If so, studies to determine the role of PPARδ in cancer-associated fibroblast formation and function would be of interest.

Unfortunately, scRNAseq lacks the sensitivity to assess the level of PPAR isoform transcripts in the remaining clusters seen in Figure 6A–C. Thus, the absence of measurable *Pparg* expression in cluster 10 is surprising. The Enrichr analysis indicated that the gene expression signature of cluster 10 cells aligned with both fibroblasts and adipocytes. PPARγ expression is well known to be markedly upregulated during adipocyte differentiation and is necessary for adipogenesis [39]. The absence of increased *Pparg* expression in this cluster indicates that differentiated adipocytes were not present. This suggests that the adipocyte features in this cluster are reflective of the presence of adipocyte progenitor cells [40] or adipocyte-derived fibroblasts [41]. It should also be noted that the absence of mature adipocytes was also by design, as our cell isolation methodology included low-speed centrifugation to remove less dense lipid-laden adipocytes from our collection. This allowed us to enhance the overall cell viability as well as to enrich our cell population for stromal fibroblasts and immune cells.

## 4. Discussion

In this report, we show that the loss of epidermal *Pparg* in mice is sufficient to induce transcriptomic changes that mimic those observed in actinic disease and SCCs. These changes include multiple changes in inflammatory signaling, chemokine expression, and immune cell recruitment. The gene set enrichment analysis revealed largely similar activation and inhibition profiles for canonical signaling pathways and diseases and biofunctions. A surprising and informative finding is that there was a switch from increased PPAR signaling in normal sun-exposed skin (SES) to a loss of PPAR signaling in AKs and cutaneous SCCs. This PPAR signaling switch correlates with a similar switch between increased *PPARG* expression in SES to reduced *PPARG* expression in malignant AK and cutaneous SCC lesions. A similar but less intense change in the pattern of *PPARA* expression was also observed. In contrast, *PPARD* expression was increased in both AKs and cutaneous SCCs. This indicates that the loss of overall PPAR signaling and an associated loss of *PPARG* and *PPARA* expression are important features that distinguish NMSC from sun-damaged skin. To our knowledge, this report is the first to demonstrate that PPAR signaling is the top commonly inhibited canonical signaling pathway in SCC development. We and others have shown that the loss of epidermal *Pparg* promotes both chemical and photocarcinogenesis in mice [9,11,12]. Thus, our data further support the idea that PPARγ acts as a potential tumor suppressor in both human and murine cutaneous SCC formation.

There are multiple potential mechanisms through which PPARγ activity could be lost in cutaneous neoplasia. These include genetic deletions or inactivating mutations. Since the germline loss of one *Pparg* allele in mice results in an increased susceptibility to chemical carcinogenesis [12], the complete loss of PPARγ activity is likely not necessary for increased tumorigenesis. The human *PPARG* gene is located at the 3p25 chromosomal locus [42]. It is therefore of interest that the loss of heterozygosity (LOH) of large portions of 3p was observed in 25% of AKs and 53% of cutaneous SCCs [43]. Another study showed that the chromosomal loss of 3p was seen in 53% of SCCs and 60% of SCCs in situ [44]. The LOH for 3p has also been described in two of five human SCC cell lines (SCC-12 and MET-1, but not SCC-13, SCL-I, or SCL-II) [45]. In addition, the LOH of chromosomal locus 3p25 is common in related head and neck SCCs (54%) [46] and laryngeal SCCs (60%) [47]. Interestingly, the LOH of 3p25 was not seen in any of the 10 cases of non-malignant laryngeal squamous metaplasia [47]. As we found that *PPARG* expression was not reduced in sun-exposed skin, this supports the idea that the loss of PPARγ expression and activity is a tumor-specific event. In contrast to *PPARG*, the LOH at chromosomal loci for the *PPARA* gene (22q) or the *PPARD* gene (6p) is infrequent (<5%) in cutaneous SCCs [43,45,48,49].

Loss-of-function somatic missense mutations of *PPARG* have also been described in cancer but are not particularly common [50]. However, as UV is a potent mutagen and UV-induced tumors have the highest reported mutation burden of human cancers [51], it is possible that somatic LOF mutations are more frequent in cutaneous SCCs.

An additional potential mechanism for the loss of PPARγ signaling in AK and SCCs includes alternative splice variants with dominant negative (dnPPARγ) activity. A number of dnPPARγ splice variants have been shown to be increased in cancer (γORF4 [52], hPPARγ1_tr_ [53], and PPARγΔ5 [54]). In mice, both TNF and LPS suppress *Pparg* expression and induce *Pparg*Δ5 splice variant expression, providing a mechanism for the loss of PPARγ activity through alternative splicing [33]. In the case of hPPARγ1_tr_, the expression of this splice variant was identified in lung SCCs but not adjacent normal tissue [53]. Thus, the upregulation of dnPPARγ variants, specifically in tumors, could also account for reduced PPARγ activity.

Reduced *PPARG* and *PPARA* expression have also been documented in some cancers through micro RNAs (miRNAs), long non-coding RNAs (lncRNAs), and promoter hypermethylation.

Using Targetscan 8.0 [55,56], 274 or 502 distinct miRNAs are predicted to target human *PPARG* or mouse *Pparg*, respectively. While the role of miRNAs in suppressing *PPARG* expression has not been adequately studied in NMSC, one study demonstrated that miR-27b, miR-130b, and miR-138 are all upregulated in colon cancer and correlate negatively with *PPARG* mRNA and protein expression [57]. In addition, the elevation of miR-374a/-128/-130b was seen in seven cases of human cutaneous SCCs relative to non-lesional skin [58]. All three of these miRNAs are predicted to bind to the human PPARG 3′-UTR, although a role in regulating *PPARG* expression was not addressed.

Like miRNAs, a number of lncRNAs have the ability to alter PPAR expression [59]. MALAT1 is a known lncRNA regulator of tumor development and is overexpressed in cutaneous SCCs [60,61]. MALAT1 is also thought to target *PPARG* [59]. In competing endogenous RNA networks (ceRNA networks), sequence homology between lncRNAs and miRNAs competes for miRNA target binding to suppress the regulatory function of miRNAs. A study of human cutaneous SCCs examined correlations between 3221 differentially expressed transcripts and 24 differentially expressed lncRNAs (DElncRNA). By incorporating known miRNA targets of the lncRNAs, they were able to predict ceRNA networks that are operational in cutaneous SCCs [62]. Of the 24 DElncRNAs that were identified, 5 were predicted to target *PPARG* (HCG18, LINC00342, HLA-F-AS1, SNX29P2, and POLR214) [62]. In addition, all five were downregulated. Thus, the loss of these lncRNAs could, in turn, promote the miRNA-induced suppression of *PPARG* expression. Consistent with this idea, *PPARG* expression was decreased in their dataset (Log2FC of −0.91071; FDR 1.73 × 10^−7^).

Another epigenetic mechanism for the potential loss of *PPARG* expression in skin cancer is promoter hypermethylation. Promoter hypermethylation of the *PPARG* gene by ubiquitin-like, containing PHD and RING finger domains, 1 (UHRF1) are associated with poor prognosis in colorectal cancer [63]. However, it has yet to be determined whether promoter methylation and the silencing of *PPARG* gene expression occur in NMSC.

Finally, recent studies indicate that PPARγ and NF-κB signaling have a complex and mutually antagonistic relationship. Studies in adipocytes, mesenchymal stem cells, and macrophages show that stimulation with LPS or TNFα act to suppress overall *PPARG/Pparg* transcript expression and target gene expression [33]. The constitutive activation of NF-κB has been described as a common feature of malignancy due to activating mutations of NF-κB transcription factors themselves or upstream regulators [64]. It is possible that the loss of *PPARG* expression or activity through any of these epigenetic or genetic mechanisms could result in a self-sustaining negative feedback cycle whereby the initial loss of PPARγ anti-inflammatory activity results in increased inflammatory cytokine production that results in a further degradation of PPARγ signaling within the tumor cells themselves or surrounding stromal cells.

The decrease in PPARα expression suggests that this transcription factor may also play a tumor suppressor role, particularly in mouse SCCs. Further studies are needed to clarify the degree to which PPARγ and PPARα activity are suppressed in NMSC and the mechanisms through which this occurs. It would therefore be of interest to determine the degree to which mice lacking epidermal *Ppara* exhibit transcriptomic changes that mirror those observed in AKs, SCCs, and *Pparg*-/-^epi^ mice.

Given that PPARγ has an important anti-inflammatory role, it is not surprising that another key finding from our analysis is that inflammatory signaling and immune cell activation are a common feature of *Pparg*-/-^epi^ mice, Aks, and SCCs. This suggests that the loss of PPARγ activity acts to promote neoplasia through its ability to induce stromal changes associated with tumorigenesis. This included a prominent accumulation of neutrophils and macrophage cells. It might be noted that mice expressing dominant negative *Pparg* (dn*Pparg*) in type II pulmonary alveolar cells stimulate the mobilization and recruitment of myeloid cells with MDSC activity [65]. Given that *Pparg*-/-^epi^ mice have a marked defect in CHS responses, it is tempting to speculate that the myeloid cell infiltrate that we observe exhibits MDSC activity. However, additional functional studies are needed to verify whether these myeloid cells represent MDSCs and contribute to the immune suppression seen in *Pparg*-/-^epi^ mice.

An interesting feature of our analysis is the increase in *PPARD/Ppard* during malignant progression. It is possible that whatever mechanism is involved in downregulating PPARγ and PPARα expression fails to elicit the downregulation of PPARδ. Alternatively, since PPAR isoforms have overlapping functions, particularly in cellular energy production [66], the increase in *PPARD/Ppard* may also serve a compensatory function to mitigate the changes that are caused by a loss of the other isoforms in human and mouse tumors. The increase in PPARδ may also promote tumor angiogenesis and progression through its effects on endothelial cells [67].

Another potential explanation for the increase in PPARδ transcripts in AKs and SCCs might be implied by our scRNAseq data from *Pparg*-/-^epi^ mice. We found that *Ppard* expression was increased in *Pparg*-/-^epi^ mice, particularly in fibroblasts expressing myofibroblast markers. It has been reported that PPARδ mediates fibroblast differentiation to profibrotic myofibroblasts by inducing the expression of TGFβ, which, in turn, induces the expression of alpha smooth muscle actin [68]. While the significance of this finding requires further studies, the presence of myofibroblasts is associated with chronic inflammation, fibrosis, and wound healing [69]. Myofibroblast markers are also associated with cancer-associated fibroblasts (CAFs) [70]. Thus, the observed increase in *PPARD*/*Ppard* expression in the tumor datasets may simply reflect an increase in myofibroblasts that are characteristic of the tumor stroma. Moreover, as PPARγ activation suppresses TGFβ expression and myofibroblast differentiation [71], this may also indicate that PPARγ and PPARδ have opposing actions in myofibroblast differentiation and fibrosis.

A weakness of our studies is that our mouse model results in the embryonic loss of *Pparg*. Thus, our studies carried out in adult mice would cause them to suffer from long-standing dermal inflammatory changes that would create a new normal homeostatic state. Thus, the proximal events that are first initiated by the loss of epidermal PPARγ cannot be assessed. If the loss of *PPARG* is a defining feature of NMSC tumor–stroma interactions, then it would be important to determine the nature of these early signals to identify potential interventional targets. While this would be difficult to achieve in our *Pparg*-/-^epi^ mice, these studies could be performed in floxed *Pparg* mice crossed with a tamoxifen-inducible Krt14-Cre transgene.

In conclusion, the loss of PPARγ expression and activity is a top feature of both *Pparg*-/-^epi^ mouse skin and AK and SCC transcriptomic datasets. A more modest reduction in PPARα expression and activity is also observed. A single-cell analysis also reveals that *Pparg*-/-^epi^ mouse skin exhibits immune cell infiltrates and myofibroblast differentiation that is indicative of a chronic inflammatory state. This genomic approach supports previous studies indicating that PPARγ is an important tumor suppressor in cutaneous carcinogenesis, leading to actinic disease and squamous cell carcinoma. Specifically, the loss of tumor cell-specific PPARγ activity may be necessary for the establishment of the stromal inflammatory microenvironment that is a hallmark of neoplastic disease in the skin.

## Figures and Tables

**Figure 1 cells-13-01356-f001:**
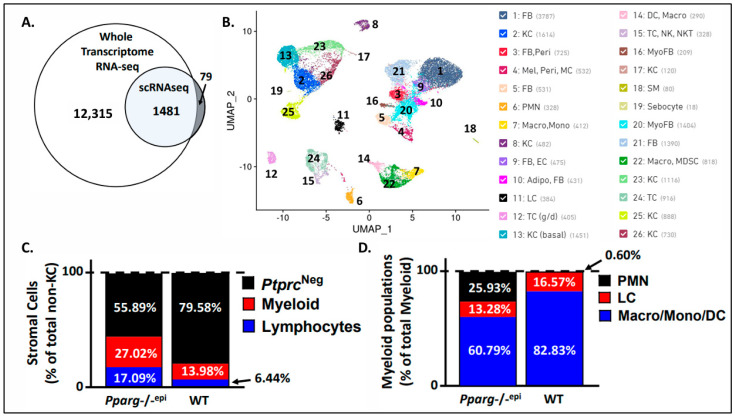
Single-cell sequencing reveals changes in the dermal cell infiltrate in *Pparg*-/-^epi^ mice. (**A**) A Venn diagram showing the overlap between differentially expressed genes (DEGs) from whole transcriptomic RNA sequencing (RNA-seq) and from the single-cell RNA sequencing (scRNAseq) of *Pparg*-/-^epi^ mouse skin. While scRNAseq suffers from reduced sensitivity, thus resulting in a much smaller number of DEGs, 94.94% of the DEGs identified by scRNAseq were also identified as DEGs by whole transcriptomic RNA-seq. (**B**) An unsupervised cell cluster analysis using uniform manifold approximation and projection (UMAP) depicts 26 cell clusters that are present in both WT and *Pparg*-/-^epi^ mouse skin. FB: fibroblasts; KC: keratinocytes; Peri: pericytes; Mel: melanocytes; MC: mast cells; PMN: polymorphonuclear cells; Macro: macrocytes; Mono: monocytes; EC: endothelial cells; Adipo: adipocytes; LC: Langerhans cells; TC (g/d): γδ T-lymphocytes; DC: dendritic cells; NK: natural killer cells; NKT: natural killer T-lymphocytes; MyoFB: myofibroblasts; SM: smooth muscle; MDSC: myeloid-derived suppressor cells. (**C**) Cytokeratin-negative stromal cells from either WT or *Pparg*-/-^epi^ skin were subdivided into CD45 (*Ptprc*)-positive and *Ptprc*-negative (*Ptprc*^Neg^) cell populations. *Ptprc*-positive immune cells were subdivided into *Cd3*-positive lymphocyte and *Cd3*-negative myeloid populations. The different populations are depicted as a percentage of total stromal cells. (**D**) The myeloid populations were further subdivided into a granulocytic (PMN) population, Langerhans cell population, and a pooled population of non-granulocytic macrophages, monocytes, and dendritic cells (Macro/Mono/DC). Each population is shown as a percentage of total myeloid cells.

**Figure 2 cells-13-01356-f002:**
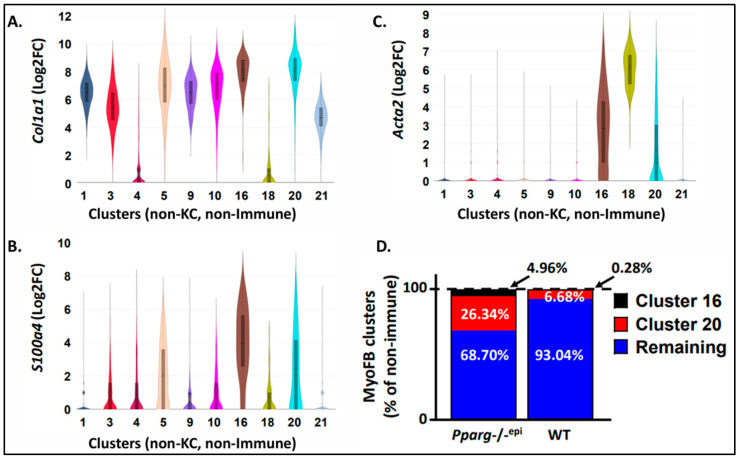
Fibroblast clusters expressing myofibroblast gene markers are increased in *Pparg*-/-^epi^ mouse skin. (**A**–**C**) Violin plots showing the log 2 fold change (Log2FC) of the following genes that were expressed in both cytokeratin-negative and CD45 (*Ptprc*)-negative stromal cell clusters: (**A**) alpha-1 type I collagen (*Col1a1*), (**B**) S100 calcium-binding protein A4 (*S100a4*), and (**C**) smooth muscle actin 2 (*Acta2*). (**D**) The percentages of cluster 16 and 20 fibroblasts that are enriched in the expression of myofibroblast (MyoFB) markers as a percentage of total non-immune stromal cells are shown for both WT and *Pparg*-/-^epi^ mouse skin.

**Figure 3 cells-13-01356-f003:**
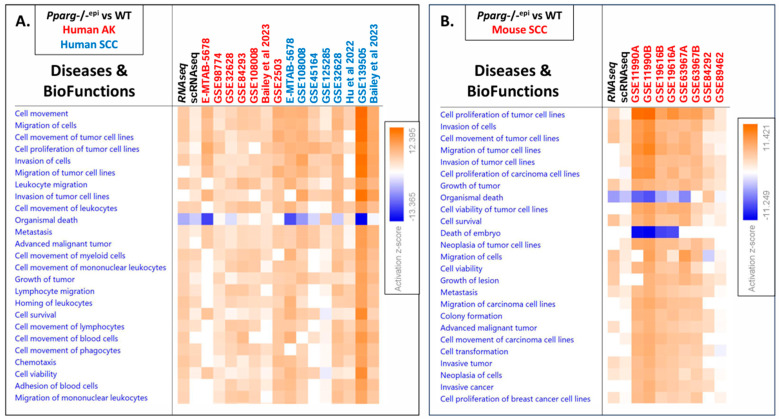
A heat map showing the top common diseases and biofunctions that are mapped to the differentially expressed genes found in *Pparg*-/-^epi^ mice, human actinic keratoses (AKs), human squamous cell carcinomas (SCCs), and mouse SCCs. Differentially expressed genes from the whole transcriptomic RNA sequencing (RNAseq) and single-cell RNA sequencing (scRNAseq) were obtained for *Pparg*-/-^epi^ relative to the WT mice. (**A**) The DEGs from these two datasets, as well as the DEGs obtained from publicly available human AK and SCC tumor databases (Appendix A), were uploaded for Qiagen’s Ingenuity Pathway Analysis. A comparison analysis was performed, and the top 25 diseases and biofunctions that were mapped to the various databases are shown and ranked by activation z-score. (**B**) A comparison heat map showing the top 25 diseases and biofunctions that are common to both the DEGs from *Pparg*-/-^epi^ mouse skin and mouse SCC transcriptomic datasets.

**Figure 4 cells-13-01356-f004:**
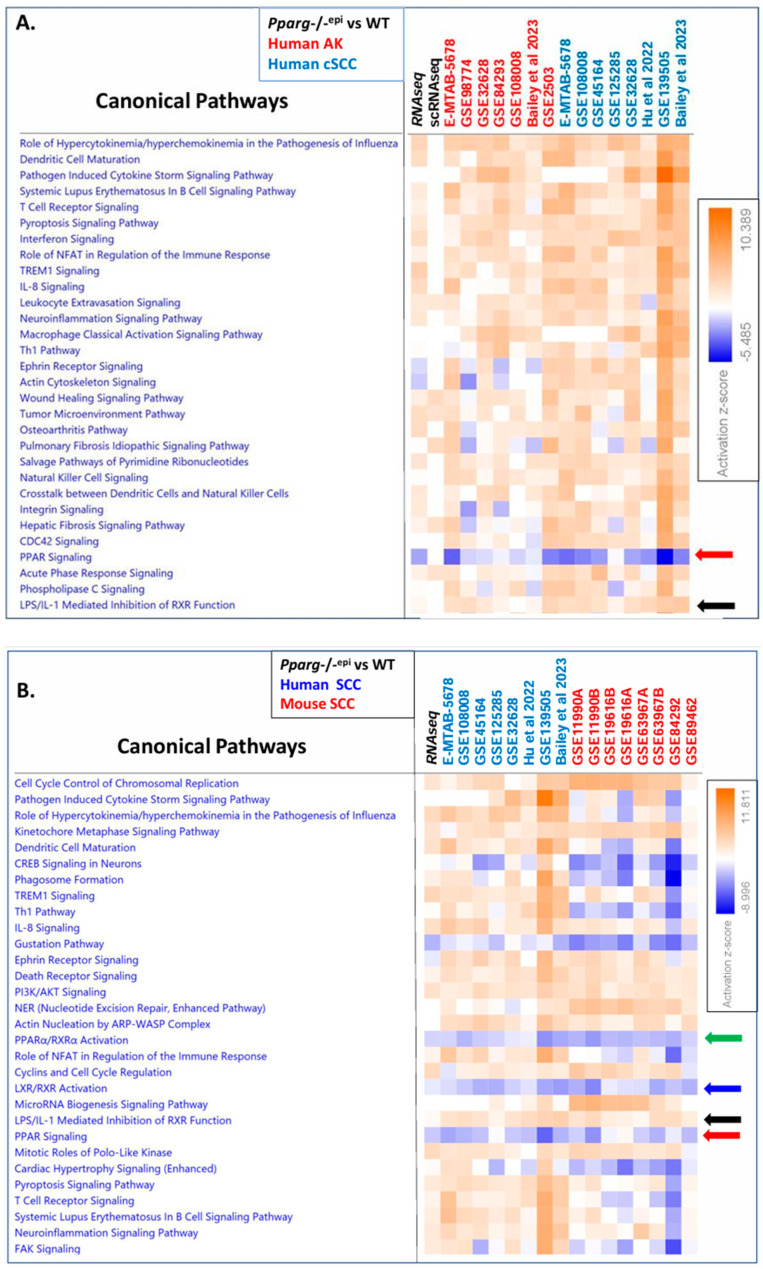
Heat maps showing top common canonical pathways that are mapped to the differentially expressed genes found in *Pparg*-/-^epi^ mice, human actinic keratoses (AKs), human squamous cell carcinomas (SCCs), and mouse SCCs. (**A**,**B**) The data analyzed in Figure 3 were further analyzed for common canonical pathways using an IPA. The canonical pathways in (**A**,**B**) are marked as follows: “PPAR Signaling” is marked by the red arrow. “LPS/IL-1 Mediated inhibition of RXR Function” is marked by the black arrow. The green arrow marks “PPARα/RXRα Activation”, and the blue arrow marks “LXR/RXR Activation”. (**A**) The top 30 canonical pathways that were mapped to the *Pparg*-/-^epi^ mouse and human AK and SCC databases are shown and ranked by activation z-score. (**B**) The top 30 canonical pathways that are common to the *Pparg*-/-^epi^ mouse skin, human SCC, and mouse SCC, transcriptomic datasets are shown (ranked by z-score).

**Figure 5 cells-13-01356-f005:**
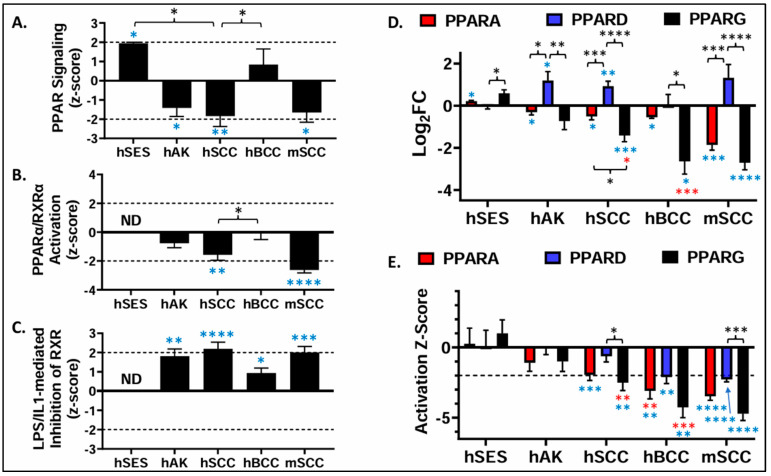
The transcriptomic analysis indicates that PPAR signaling is inhibited, while PPARγ (and PPARα) expression and activity are suppressed in human AKs, human SCCs, and mouse SCCs. (**A**–**C**) Following GSEA, activation z-scores were obtained for each of the different datasets for the following canonical pathways: (**A**) PPAR signaling, (**B**) PPARα/RXRα activation, and (**C**) LPS/IL1-mediated inhibition of RXR. The plots depict the mean and SEM of the activation z-scores for the different canonical pathways. The hashmark lines represent the cutoff for the predicted activation (2.0 activation z-score) or predicted inhibition (–2.0) of the respective canonical pathway. For the hSES dataset, no predictive z-score was provided for these canonical pathways (ND = no data). (**D**) For each dataset, the mRNA expression for each of the three different PPAR isoforms was obtained as the Log_2_ fold change (Log2FC). The data shown are the mean and SEM of the Log_2_FC. (**E**) After uploading the DEG data from each tumor or hSES dataset for the IPA, activation z-scores were obtained for PPARα, PPARδ and PPARγ. The data shown are the mean and SEM of the activation z-score for each PPAR isoform. The hashmark line represents the cutoff for the predicted inhibition (−2.0) of the respective PPAR isoform. Blue asterisks = significantly different from zero. One sample T-test. Red asterisks = significantly different from hSES expression (1-way ANOVA). Black asterisks = significantly different from each other (1-way ANOVA). *, *p* < 0.05; **, *p* < 0.01; ***, *p* < 0.001; ****, *p* < 0.0001.

**Figure 6 cells-13-01356-f006:**
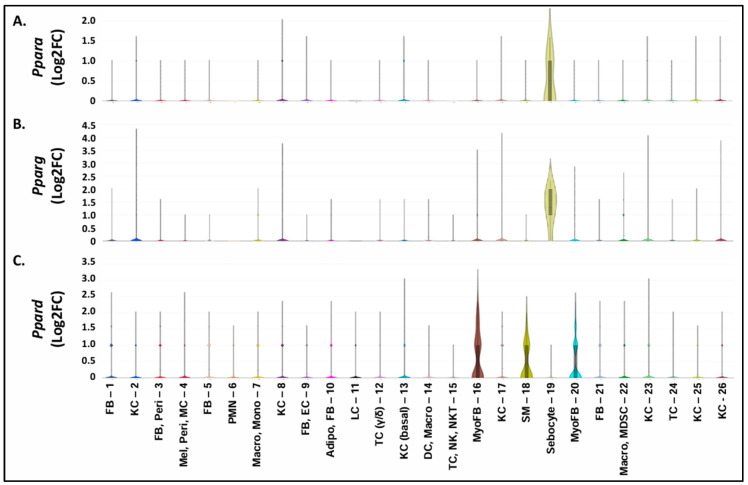
Cell clusters with a high level of expression of the three PPAR isoforms. (**A**–**C**) The expressions of *Ppara* (**A**), *Pparg* (**B**), and *Ppard* (**C**) are shown for all of the individual cell clusters obtained following the scRNAseq of skin cells (combined cells isolated from both WT and *Pparg*-/-^epi^ mouse skin). Both *Ppara* and *Pparg* expression were highly enriched in sebocyte cluster 19. In contrast, *Ppard* expression was enriched in myofibroblasts (clusters 16 and 20) and smooth muscle cells (cluster 18).

**Table 1 cells-13-01356-t001:** The top 20 disease or biofunction terms that match with the DEG dataset from *Pparg*-/-^epi^ mouse skin relative to WT skin (sorted by z-score).

Disease or Biofunction Annotation	*p*-Value	Activation z-Score
Chemotaxis of leukocytes	1.61 × 10^−12^	5.052
Homing of leukocytes	8.69 × 10^−14^	4.724
Cell movement of tumor cell lines	3.36 × 10^−11^	4.686
Chemotaxis	2.68 × 10^−18^	4.644
Migration of cells	1.13 × 10^−30^	4.623
Homing of blood cells	4.72 × 10^−14^	4.574
Cell survival	2.9 × 10^−15^	4.568
Leukocyte migration	2.98 × 10^−29^	4.468
Cell movement	2.97 × 10^−37^	4.464
Cell movement of blood cells	1.82 × 10^−29^	4.36
Invasion of cells	9.29 × 10^−15^	4.359
Homing of cells	3.91 × 10^−21^	4.249
Inflammatory response	9.14 × 10^−20^	4.174
Cell movement of myeloid cells	2.13 × 10^−18^	4.171
Recruitment of myeloid cells	1.19 × 10^−13^	4.121
Cell viability	1.77 × 10^−14^	4.086
Recruitment of blood cells	3.44 × 10^−19^	4.07
Cell movement of phagocytes	5.77 × 10^−19^	4.029
Growth of lesion	5.78 × 10^−30^	3.992
Growth of tumor	1.15 × 10^−29^	3.942

## Data Availability

The raw data supporting the conclusions of this article will be made available by the corresponding author upon request. Some of the data in this report were derived from resources available in the public domain [see Appendix A for public dataset sources].

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
