# Peer review of "The Loss of PPARγ Expression and Signaling Is a Key Feature of Cutaneous Actinic Disease and Squamous Cell Carcinoma: Association with Tumor Stromal Inflammation"

_cells, 2024, doi:10.3390/cells13161356_

Round 1

Reviewer 1 Report

Comments and Suggestions for Authors

Generally, this is an interesting manuscript with that makes useful contributions to the literature as a whole by investigating whether the depletion of PPARgamma in K14 expressing tissue, and subsequent inflammation in a C57/Bl6 mouse KO model has relevance to human skin conditions such as psoriasis, AK, SCC, and BCC.

However, this MS is a sort of "hodge-podge" of data base query results and  information that lacks necessary detail in places and would benefit from significant editing to improve clarity of objectives, methods, and findings.

The methods are extremely brief.  Sufficient info that enables the paper to stand alone from the authors' previous publications would be helpful.

The mice used appear to have been backcrossed into a C57/Bl6 background for the prior studies.  Why was this strain that is less susceptible to an inflammatory response/tumor promotion used for these experiments? How does background this influence the subsequent comparative analyses to public data?

The final conclusion in the abstract is rather strong.  The supporting evidence is sufficient for this statement as a hypothesis to be tested, but since the findings are from evaluating data that happened to be available, the conclusion may be somewhat overstated.

Section 3.1.  This section would benefit from rewrite for clarity.  The authors analyzed a previously published transcriptomic dataset in IPA.  Lines 197-198 - does the reference to DEGs refer to genes differentially expressed between the Ppargamma KO mice and their matched controls described in the previous publication?

Line 206.  The pathways data supplied by IPA and others reflects data deposited.  In these types of analysis, "Cancer" is always overrepresented.  The results reflect what is in the database.  Database queries are useful for revealing insights and hypothesis testing, but in and of themselves, they are limited in their utility.  Differences in p-values between psoriasis and atopic dermatitis studies vs cancer is almost certainly a statistical artifact.

Line 213.  The leap from inflammation to tumor/cancer vs malignancy, especially in B6 mice is a big jump, so the lower z score for malignancy is not surprising.  Did you query the database as a whole or limit to other epithelial-tissue studies?

Section 3.2 and single cell methods.  What was the biological and statistical rationale for 2 mice, one female and one male?  What was the rationale for final pooling before sequencing, including the 1:10 dermis to epidermis?

How does histological findings from the previous study corroborate the infiltration of neutrophils in the ko mice?  What state were the two mice in (e.g. hair growth phase, epithelial inflammation, etc) compared to WT in this experiment?

Regarding BCCs - generally there is a relationship between inflammation, AK, and SCCs in human and mouse NMSCs. BCCs are a different tumor type in humans and certainly in mouse carcinogenesis models - again the discordance is not at all surprising  - likely not tumor heterogeneity as much as the fact that different tumor types have different transcriptome profiles.

3.6 this section is interesting but the shift to sun exposed vs. not is a significant departure from mouse studies.

Discussion: line 579 i think this should specify "in mice."

line 595 These data support a role for PPARg as a potential tumor suppressor.  This biased database query is not sufficient for stronger conclusions, but does support a strong hypothesis.

Line 718 - Not sure that your approach to fishing the database should be labled as "non-biased."

Other comments:

Line 162. SoupX was used to remove the ambient RNA _sequences_ from the data.  Also is "ambient" the best word to use here?

Line 173. Mitochondrial is misspelled.

Author Response

1.) The methods are extremely brief.  Sufficient info that enables the paper to stand alone from the authors' previous publications would be helpful.

Response:  We agree that it is difficult to balance the need for brevity and an adequate description of the methodologies utilized in a publication.  To help the reader determine how this new analysis differs from the previously published study [Int J Mol Sci 2021, 22, 8634], we have added a few lines describing how the transcriptomic dataset was analyzed for this previous study. “Sequencing was done on RNA obtained from the skin of 6 mice per genotype [6]. In this previously published study, we characterized how changes in gene expression correlated with the phenotypic changes that we observed in Pparg-/-epi mice. This included spontaneous inflammatory skin lesions, an asebia phenotype, and epidermal permeability defects.

2.) The mice used appear to have been backcrossed into a C57/Bl6 background for the prior studies.  Why was this strain that is less susceptible to an inflammatory response/tumor promotion used for these experiments? How does background this influence the subsequent comparative analyses to public data?

Response:  We agree that the C57BL6/J background is resistant to both UV and chemical carcinogenesis relative to mice such as the SKH1 outbred strain.  However, loss of Pparg in K14 expressing cells of both SKH1 and C57BL6 mice result in a similar suppression of CHS responses in the absence of any UV treatments. Our goal was to examine the immune cell types and cytokines/chemokines that are altered that could explain the immunosuppressed phenotype.  This immune suppression would likely be relevant to cancer development.  In addition, a disadvantage of the SKH1 background is that they are an outbred strain. For the current study, we felt that the use of the congenic C57BL6 background is preferable due to the potential confounding variable that a non-congenic strain would introduce to NGS studies. 

3.) The final conclusion in the abstract is rather strong.  The supporting evidence is sufficient for this statement as a hypothesis to be tested, but since the findings are from evaluating data that happened to be available, the conclusion may be somewhat overstated.

Response: The final conclusion in the abstract has been changed as follows: The original sentence has been softened to the following statement -  “In conclusion, these data provide strong evidence that PPARγ and possibly PPARα represent key tumor suppressors by acting as master inhibitors of the inflammatory changes found in AKs and SCCs.”

4.) Section 3.1.  This section would benefit from rewrite for clarity.  The authors analyzed a previously published transcriptomic dataset in IPA.  Lines 197-198 - does the reference to DEGs refer to genes differentially expressed between the Ppargamma KO mice and their matched controls described in the previous publication?

Response: Yes, the DEGs referred to those transcripts that were differentially expressed in the Pparg-/-epi mouse skin relative to WT mouse skin.  We have edited section 3.1 to make this clear. 

5.) Line 206.  The pathways data supplied by IPA and others reflects data deposited.  In these types of analysis, "Cancer" is always overrepresented.  The results reflect what is in the database.  Database queries are useful for revealing insights and hypothesis testing, but in and of themselves, they are limited in their utility.  Differences in p-values between psoriasis and atopic dermatitis studies vs cancer is almost certainly a statistical artifact.

Response:  We are very appreciative of this reviewer’s insight.  We performed a similar IPA analysis on using publicly available transcriptomic datasets for atopic dermatitis and psoriasis (DEGs were obtained for lesional versus nonlesional psoriatic or atopic dermatitis patients).  As with our Pparg-/-epi dataset, a very similar statistical linkage to the same cancer disease and biofunctions was obtained.  While this could also reflect to some extent the fact that inflammation is a hallmark of the cancer microenvironment, the reviewer is correct in that the statistical significance could simply reflect large numbers of cancer datasets that result in statistical overweighting. Thus, we have removed Table 1A as well as the associated results section discussion from the revised manuscript. 

6.) Line 213.  The leap from inflammation to tumor/cancer vs malignancy, especially in B6 mice is a big jump, so the lower z score for malignancy is not surprising.  Did you query the database as a whole or limit to other epithelial-tissue studies?

Response: As noted above for comment #5, we have removed table 1A as well as the associated discussion from the revised manuscript.  However, in answer to your question, the query was done using the database as a whole. 

7.) Section 3.2 and single cell methods.  What was the biological and statistical rationale for 2 mice, one female and one male?  What was the rationale for final pooling before sequencing, including the 1:10 dermis to epidermis?

Response:  Biological and statistical rationale for 2 mice: As noted above, Pparg-/-epi mice have a profound loss of contact hypersensitivity responses, (see our publication in Oncotarget. 2017 Nov 17; 8(58): 98184–98199).  Our initial goal for the single cell sequencing studies was to determine the differences in immune cell populations that could provide some insight into the cells involved in this immune suppressed phenotype.  Since the differences in the CHS responses are so dramatic between the WT and Pparg-/-epi mice, and the variances within each genotype are so small, we felt that two biological replicates was sufficient to provide a meaningful analysis of the dermal immune cell microenvironment.  In addition, in Fig 1A, we show very good correlation between gene transcripts that are differentially expressed in the single cell RNA sequencing studies and the earlier RNA sequencing experiments.  Pooling before sequencing, including the 1:10 dermis to epidermis:  The pooling of samples was done as cells from both male and female mice can be easily discriminated based on sex-specific gene expression.  For the pooling of the epidermal cells, our primary goal was identifying the stromal inflammatory cells that could account for the immune suppression and inflammatory phenotype that we see in Pparg-/-epi mice.  This is particularly important for identifying rather infrequent immune cells, such as regulatory T-cells. Since the sequencing run was limited in the number of cells that could be sequenced, we wished to limit the number of keratinocytes to allow for a greater number of immune cells that could be sequenced.  As a whole skin cell prep would be heavily weighted towards keratinocytes, we performed the epidermal cell isolation separately and added these cells at a reduced amount. 

8.) How does histological findings from the previous study corroborate the infiltration of neutrophils in the ko mice?  What state were the two mice in (e.g. hair growth phase, epithelial inflammation, etc) compared to WT in this experiment?

Response:  We have done some immunolabeling studies in the WT and Pparg-/-epi mice that corroborate the single cell data that shows an increase in myeloid cells. However, we did not use a neutrophil-specific antibody.  These studies are ongoing and will be published in a follow-up manuscript that will examine how myeloid and lymphoid populations contribute to the CHS defect seen in Pparg-/-epi mice.  Cells were isolated from areas of the skin in which the hair follicles were in telogen phase.  We have added this information to the methods section. 

9.) Regarding BCCs - generally there is a relationship between inflammation, AK, and SCCs in human and mouse NMSCs. BCCs are a different tumor type in humans and certainly in mouse carcinogenesis models - again the discordance is not at all surprising  - likely not tumor heterogeneity as much as the fact that different tumor types have different transcriptome profiles.

Response:  We agree that mice are a poor model for BCC and that the lack of correlation between the Pparg-/-epi mice and the BCC datasets is not particularly surprising.  We have added the following sentence to the results section that is discussing Fig S3 (lines 337-340): “As BCCs are not found in mice lacking disruptions in patched signaling, it is not surprising that there was limited correlation between the diseases and biofunction annotations that were common to both the Pparg-/-epi dataset and the BCC datasets.” However, in lines 340 – 344 of the revised manuscript, our comment regarding the discordance between the diseases and biofunctions linked to BCC datasets referred to the lack of agreement between BCC datasets, not the lack of agreement between the Pparg-/-epi and BCC datasets. 

10.) 3.6 this section is interesting but the shift to sun exposed vs. not is a significant departure from mouse studies.

Response:  We agree that this finding is not directly addressed by our mouse findings. However, this data does provide additional evidence that loss of PPARγ expression and signaling correlates with malignancy and is not simply a feature of sun-damaged skin.  It adds to the other evidence that supports a role for PPARγ (and possibly PPARα) as tumor suppressing signals.  Future studies in humans and mice are certainly necessary to provide additional supporting evidence as well as mechanistic details. 

11.) Discussion: line 579 i think this should specify "in mice."

Response: Agree. This change has been made. (now line 594 in the revised manuscript).   

12.) line 595 These data support a role for PPARg as a potential tumor suppressor.  This biased database query is not sufficient for stronger conclusions, but does support a strong hypothesis.

Response:  Agree.  We have softened the statement to indicate that the data is supportive of a potential tumor suppressor role in SCC: “Thus, our data further supports the idea that PPARγ acts as a potential tumor suppressor in both human and murine cutaneous SCC formation.”  (now lines 610-612 in the revised manuscript).  

13.) Line 718 - Not sure that your approach to fishing the database should be labled as "non-biased."

Response: We have removed the non-biased label from this sentence. (now line 736 in the revised manuscript).

Other comments:

14.) Line 162. SoupX was used to remove the ambient RNA _sequences_ from the data.  Also is "ambient" the best word to use here?

Response: This was the terminology used by the designers of SoupX in reference 18: Young, M.D.; Behjati, S. SoupX removes ambient RNA contamination from droplet-based single-cell RNA sequencing data. Gigascience 2020, 9, doi:10.1093/gigascience/giaa151.  However, the point is the removal of contaminating cell-free RNA.  Thus, we have changed the test to read cell-free contaminating RNA.

15.) Line 173. Mitochondrial is misspelled.

Response: Thanks, this has been corrected (line 178 of revised manuscript).

Reviewer 2 Report

Comments and Suggestions for Authors

I reviewed the manuscript “Loss of PPARγ expression and signaling is a key feature of cutaneous actinic disease and squamous cell carcinoma: Association with tumor stromal inflammation” submitted by Konger RL and colleagues. In this manuscript, the authors report on transcriptomic analyses of tissue (epidermal)-specific PPARγ deficient mice to uncover the roles of PPARγ in the development of skin lesions and cancer.

They report several pathways linked to the inflammatory response, neutrophil infiltration, and stromal cell proliferation being activated in PPARγ deficient mice, indicating this nuclear receptor's strong causal role in cutaneous actinic disease and carcinoma. They also report correlations of mouse transcriptomic results with publicly available human datasets.

Overall, the study is well-designed. The manuscript is well organized and conclusions are supported by the results. The limitation of the study is that is entirely based on comparative analyses of gene expression data. Although this is very informative, validating the identified pathways in human biopsies or mouse tumors would improve the manuscript and is warranted. 

Author Response

Comment 1.  Although this is very informative, validating the identified pathways in human biopsies or mouse tumors would improve the manuscript and is warranted. 

Response: We agree that additional human or mouse data would strengthen the manuscript.   One obvious approach would be to examine PPARγ expression in mouse SCCs and human AK/SCCs by immunolabeling.  However, we have tried 2-3 different PPARγ antibodies that are reported to detect both human and mouse PPARγ by IHC. Unfortunately, all of them show non-specific binding when we utilize our knockout mouse skin as controls.  Thus, these studies would require additional time to work out, assuming that we can find suitable reagents.  Alternatively, we could provide additional details of the immune cell populations found in Pparg-/-epi mouse skin and whether these cells play a role in the marked defect in contact hypersensitivity that we have documented in these mice.  These studies are ongoing. However, once completed, would be too extensive to include in this current manuscript.  We are also planning on performing experiments in mouse SCC cell lines in which the Pparg gene is deleted/knockdown or overexpressed, then determine whether this alters the growth, invasiveness, and immune microenvironment following transplantation into syngeneic mice.  Again, these studies would be complicated, would be too extensive to include in this current manuscript, and would unduly delay the publication of these current results.  In conclusion, the studies that we have ongoing will take some time to complete, would delay the release of this current interesting information, or would be too extensive to place in a single manuscript. 

Reviewer 3 Report

Comments and Suggestions for Authors

The manuscript is well written and presents the results of analysing transcriptomic changes in epidermal PPARγ-deficient mice (Pparg-/-epi) using single cell sequencing. The experiments were well designed and appropriate methods were used to analyse the scRNAseq data. All references used are appropriate and used correctly.

In my opinion, the manuscript can be accepted for publication in this form after minor revision.

Minor revisions:

Subchapter: 2.4. single cell sequencing and library preparation protocol

It should be added how many reads were made during single-cell RNA sequencing.

Author Response

Comment 1.  Subchapter: 2.4. single cell sequencing and library preparation protocol: It should be added how many reads were made during single-cell RNA sequencing.

Response:  The data in Figures S1 & S2 include the mean reads/cell, the median genes per cell, and other sequencing related QC data.    

Round 2

Reviewer 1 Report

Comments and Suggestions for Authors

Thank you for the thoughtful and complete responses to prior questions and comments.

The edits to the text have improved clarity of the manuscript.